# Cortical modulation of sensory flow during active touch in the rat whisker system

Shubhodeep Chakrabarti [1,2] & Cornelius Schwarz [1,2]

Sensory gating, where responses to stimuli during sensor motion are reduced in amplitude, is a hallmark of active sensing systems. In the rodent whisker system, sensory gating has been described only at the thalamic and cortical stages of sensory processing. However, does sensory gating originate at an even earlier synaptic level? Most importantly, is sensory gating under top-down or bottom-up control? To address these questions, we used an active touch task in behaving rodents while recording from the trigeminal sensory nuclei. First, we show that sensory gating occurs in the brainstem at the first synaptic level. Second, we demonstrate that sensory gating is pathway-specific, present in the lemniscal but not in the extralemniscal stream. Third, using cortical lesions resulting in the complete abolition of sensory gating, we demonstrate its cortical dependence. Fourth, we show accompanying decreases in whisking-related activity, which could be the putative gating signal.

[1] Department of Cognitive Neurology, Hertie Institute of Clinical Brain Research, Eberhard Karls University of Tübingen, 72076 Tübingen, Germany. [2] Systems Neurophysiology, Werner Reichardt Center for Integrative Neuroscience, 72076 Tübingen, Germany. Correspondence and requests for materials should be addressed to S.C. (email: shubhodeep.chakrabarti@cin.uni-tuebingen.de)

Animals actively sample their environment to acquire crucial information from their surroundings for efficient foraging and navigation[1–3]. An interesting property shared across a wide variety of active sensing systems is sensory gating in which sensory responses acquired during sensor motion are reduced in amplitude[4–14].

An excellent and highly efficient example of an active sensing system is the rodent whisker system. The vibrissae on the snout act as tactile sensors, actively sampling the environment (whisking) and palpating surrounding objects to extract tactile cues essential for guiding the animal's subsequent behavior[15,16]. The whisker follicle is innervated by primary afferents conveying sensory responses to several brainstem trigeminal sensory nuclei (TSN)[17–19]. Here second-order neurons carrying ascending sensory information diverge into at least four separate parallel processing pathways. The three best described of these, the lemniscal, paralemniscal and extralemniscal pathways arise from neurons in the principal (Pr5), the rostral and the caudal interpolaris (Sp5i) nuclei, respectively[20–24] and project onto third-order neurons in different parts of the somatosensory thalamus, which in turn project to the different regions of sensorimotor cortex[24–26]. It has been suggested that these pathways encode different features viz., touch and whisking[23].

Recent work from our group, as well as others, show that the TSN, in turn, all receive robust descending projections from primary (S1) and secondary (S2) somatosensory as well as posterior parietal cortices, thus closing the loop[27,28]. Importantly, no direct projections from motor cortex (M1) to the TSN were labeled. What are the putative signals conveyed to the TSN from the parietal cortices via these profuse corticofugal pathways and what functions do they serve? Sensory gating has been widely reported in rodents with thalamic and cortical sensory responses being attenuated during whisking[8–10,14]. However, relatively little is known about the origins of sensory gating, its causal mechanisms, or its site of action. First, it is unclear where exactly ascending sensory signals are initially gated. In the primate, gating occurs in the spinal cord, at the first synaptic level[11,12]. An earlier report has suggested that whisker-related sensory signals are similarly gated in the TSN[10], which are the analogous structures in the rodent vibrissal system, a hypothesis that remains to be tested. Second, it is not known whether gating is a global phenomenon affecting all ascending pathways, or if it is a highly refined process that selectively acts on specific sensory processing streams as in the monkey somatosensory system[29]. Third, and most importantly, it is still unclear if sensory gating is propagated in a bottom-up manner or whether it is a top-down phenomenon with active cortical involvement. Reports of sensory gating during passive limb movement[4,7], although contested[5], and its disappearance upon circumventing peripheral stimulation[10] seem to suggest the former. However, persistence of sensory gating upon deafferentation[4,9] and inhibition of sensory information flow by cortical stimulation[28,30,31] implicate cortical involvement. The crucial experiment to distinguish between these two possibilities would involve the quantification of sensory gating in the absence of cortical drive and has never been attempted.

In this study, we used an active touch task in rats in combination with chronic neural recordings in the TSN to determine the site of action of sensory gating, its effect on the lemniscal and the extralemniscal processing streams, and its cortical dependence. First, we showed the occurrence of sensory gating in Pr5 neurons of the whisker system. Given the very low number of inhibitory interneurons in the Pr5, these most likely represent second-order neurons[32]. Second, we demonstrate that sensory gating in the whisker system is a pathway-specific phenomenon affecting the lemniscal, but not the extralemniscal processing

stream. Third, using lesions of the whisker representations of S1 and S2, the two main sources of corticofugal input to the TSN[27,28,33], we determine the critical role of sensorimotor cortex in sensory gating, thus supporting the top-down hypothesis of sensory gating. Our data are the first to unambiguously show the critical dependence of sensory gating on top-down, corticofugal modulation at the earliest stage of sensory processing.

## Results

**Sensory gating occurs in the brainstem.** Animals actively whisked against a moving object (Fig. 1a) to obtain water reward. Contacts with the object during both non-whisking and whisking epochs were rewarded (Fig. 1b). Animals learnt the task in a few sessions and performed between 300 and 500 trials (~45 min) each day. Pre-contact velocities were used to classify whisker contacts as whisking and non-whisking contacts (Fig. 1c, d). Whisking contacts had significantly stronger decelerations around object contact as can be expected from the relative motion of the whisker and the object (Fig. 1d)[9]. Only trials contained in overlapping parts of minimum deceleration distributions of whisking and non-whisking trials were selected for further analyses. The introduction of this matched pre-selection of trials did not change any of the evoked effects mentioned in this manuscript (Supplementary Fig 1).

As sensory gating has previously been shown at the thalamic and cortical stages of the lemniscal pathway[8–10,14], we first determined whether it originates even earlier by using chronic electrode arrays implanted in the Pr5. An example of sensory responses to whisker deflections during whisking and non-whisking in a Pr5 single unit is shown in Fig. 2a. Pr5 units were almost always single-whisker (SW)-responsive and responded with a short latency (<5 ms). The response duration was long with firing rates reaching baseline levels only ~30 ms after object contact. Most importantly, the peak responses during non-whisking (dark blue) were larger in amplitude than those during whisking epochs (light blue), showing the occurrence of sensory gating in the Pr5, as has been hypothesized previously[10]. This particular example showed two response peaks, the second of which showed gating. Ten of the 23 neurons recorded in Pr5 in the intact animals showed the presence of such double peaks in their peri-stimulus time histograms (PSTHs) (Supplementary Fig 2). In these neurons, the first peak occurred within the first 10 ms and was consistent in both whisking and non-whisking trials, whereas the second peak occurred between 6 and 25 ms and had greater temporal variability (Supplementary Fig 2A). Quantitative analysis, however, showed that the strength of gating was in fact similar for both these peaks (Supplementary Fig 2B, $p = 0.56$, Wilcoxon's signed-rank test).

Population data normalized to the mean baseline firing rate (10 ms window starting 20 ms before touch) is shown in Fig. 2b. The magnitude of sensory gating in each unit was computed as the ratio between the peak evoked responses (10 ms after contact) in whisking and non-whisking epochs ($W_{max}/NW_{max}$) and are plotted on the abscissa in Fig. 2c. Similar ratios between non-normalized, mean baseline firing rates ($\overline{W}/\overline{NW}$), which provided a measure of whisking-related increases in firing rate, are shown on the ordinate. All but two of the evoked ratios were <1 ($\mu$: 0.7, $\sigma$: 0.2) indicating a statistically significant effect of sensory gating resulting in stronger responses during the non-whisking epoch (effect size $W_{max}$ vs. $NW_{max}$: 0.65, $n = 23$, $p < 0.01$, Wilcoxon's signed-rank test). In contrast, the baseline ratios were all positive except one ($\mu$: 3.8, $\sigma$: 4.4, effect size $\overline{W}$ vs. $\overline{NW}$: 0.29, $n = 23$, $p < 0.01$, Wilcoxon's signed-rank test) demonstrating a significant increase in firing rate during the whisking epoch, as has been reported previously[34].

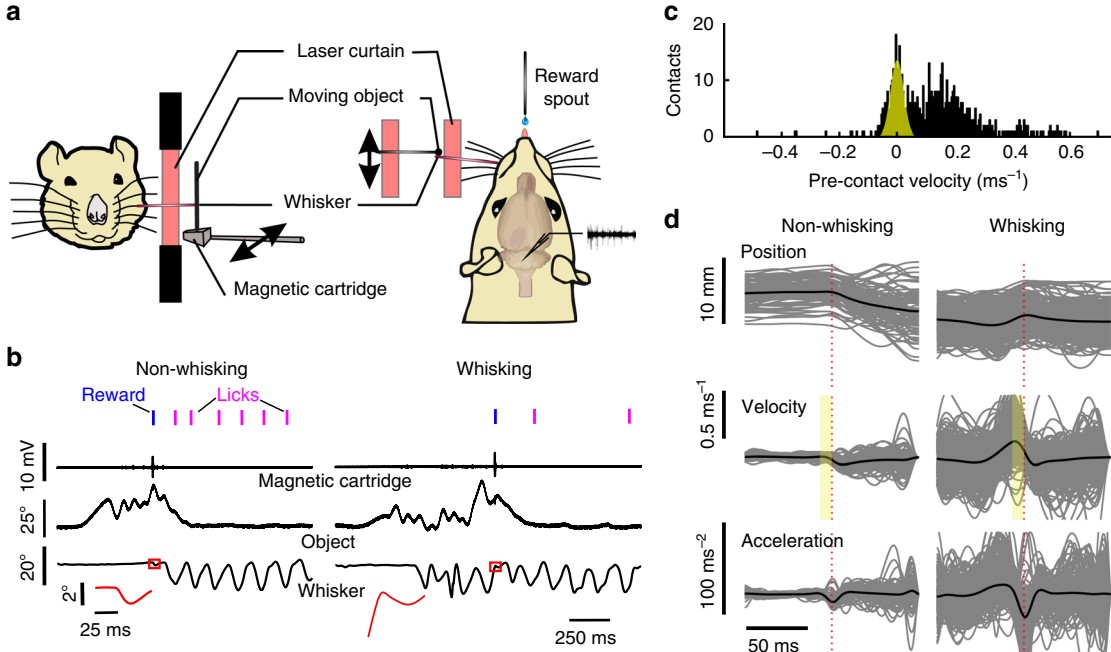

**Fig. 1** Active touch task for measuring sensory gating. **a** Schematic showing behavioral setup. The head-fixed animal was positioned next to an object, which was moved in a rostro-caudal trajectory parallel to the animal's snout. Whisker contacts were rewarded with a drop of water from a reward spout. Whisker movements and object trajectories were monitored using a laser curtain, whereas whisker contacts were detected by a vibration-sensitive magnetic cartridge. **b** Examples of whisker contacts during non-whisking (left) and whisking (right) epochs. Blue lines show rewarded licks, whereas magenta lines denote consummatory licks. Insets (red squares) show whisker trajectories around object contact, which are shown enlarged in red. **c** Pre-contact velocities (10 ms preceding contact) were binned to yield bimodal distributions. The peak around zero was fitted with a Gaussian function and pre-contact velocities falling within double the standard deviation defined as non-whisking (yellow shaded area). **d** Contact-triggered traces of whisker position, velocity, and acceleration shown for non-whisking (left) and whisking (right) epochs

**Sensory gating is pathway-specific**. Having shown that sensory gating occurs in the Pr5, we next determined whether this was also true for the extralemniscal pathway by recording from the caudal sector of the Sp5i nucleus. The Sp5i contains two groups of neurons, multi-whisker glutamatergic neurons, which project to thalamus and SW GABA- or glycinergic neurons that project to other TSN, including the Pr5[20,32,35].

First, a recent study in the primate[29] has shown that movement gates sensory activity only in specific ascending channels. Whether sensory gating in the whisker system is similarly pathway-specific is unknown and recording from the Sp5i would clarify this. Second, the inhibitory intra-trigeminal projections to the Pr5 and other TSN arising from the SW Sp5i interneurons have been hypothesized to be instrumental in sensory gating[28,32] via a cascade of inhibitory projections (from Sp5c to Sp5i further to Pr5). Matching this view, it has been shown that Sp5i blockade with bupivicaine abolishes sensory gating[10]. In the primate sensorimotor system, GABAergic interneurons are thought to be recruited in a similar way for presynaptic inhibition at the primary afferents synapse in the spinal cord[11,36]. Recording from the Sp5i interneurons would therefore test this hypothesis, as sensory responses in these neurons should be stronger during the whisking epoch if they are indeed involved in increased inhibition of Pr5 activity.

Figure 2d, e shows population PSTHs of sensory responses of all units (green) and the SW-responsive putative interneurons (SW, magenta, exclusively single units) recorded from the caudal Sp5i during whisking (light color) and non-whisking (dark color) epochs. First, the whisking and non-whisking PSTHs were largely congruent for the entire Sp5i neuronal population (Fig. 2d) as well as the putative interneurons (Fig. 2e) unlike the situation in Pr5 (cf. Fig. 2b). This was also reflected in the distributions of

$W_{max}$ vs. $NW_{max}$ ratios shown in Fig. 2f, which were centered around 1 ($\mu$: 1.1, $\sigma$: 0.5) with nonsignificant effect sizes for both the putative interneurons (effect size: 0.45, $n = 17$, $p = 0.36$, Wilcoxon's signed-rank test) as well as the entire population (effect size: 0.49, $n = 64$, $p = 0.98$, Wilcoxon's signed-rank test), respectively. Second, the effect of whisking on the baseline firing rates, which were systematically elevated for the Pr5 units (Fig. 2c) during whisking was absent in Sp5i interneurons as seen from the magenta data points ($\overline{W}$ vs. $\overline{NW}$ ratios, $\mu$: 1.1, $\sigma$: 0.5) along the y-axis in Fig. 2f (effect size: 0.45, $n = 17$, $p = 0.92$, Wilcoxon's signed-rank test). Units having a baseline firing rate of zero during non-whisking epochs are shown at the top of the ordinate as having infinite $\overline{W}$ vs. $\overline{NW}$ ratios. A weak but significant effect of whisking on baseline firing rates was, however, observed for the entire population of Sp5i units shown by the green data points on the same graph (effect size: 0.48, $n = 64$, $p < 0.01$, Wilcoxon's signed-rank test). Thus, both groups of units failed to show any effect of sensory gating, although the entire population did show a weak effect of whisking onset on firing rate. Our data thus demonstrate that sensory gating in the rodent whisker system is a remarkably refined, pathway-specific phenomenon in striking similarity with the primate somatosensory system[29]. Further, our data argue against an essential role of the Sp5c-Sp5i-Pr5 inhibitory cascade as the mediator of sensory gating[28] as we failed to see systematic increases in evoked responses during whisking in the putative interneuron population.

**Sensory gating is cortex-dependent**. Next, we tested the hypothesis that sensory gating depends on corticofugal modulation by placing large aspiration lesions over the S1 and S2 whisker representations, which had been mapped out in each animal

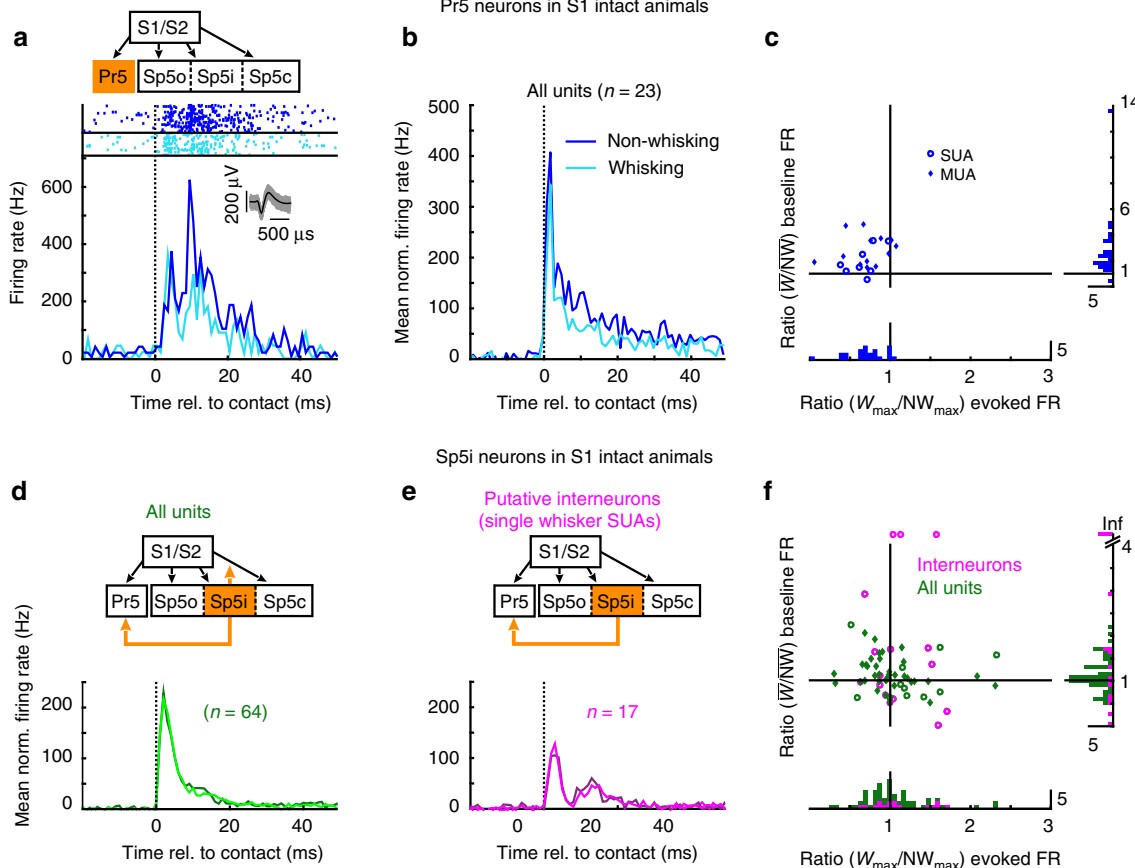

**Fig. 2** Sensory gating occurs in the brainstem and is pathway-specific. **a** A representative example of a single unit recorded from the nucleus principalis (Pr5) with rasters and peri-stimulus time histograms (PSTHs) showing contact-evoked responses during whisking (light blue) and non-whisking (dark blue) epochs (bin width: 1 ms). Inset shows the first 1000 spike waveforms (gray) with superimposed median (black). **b** Population PSTHs (mean across units) of all units, single and multi, recorded from Pr5 normalized to baseline firing rates (20–10 ms pre-contact). Vertical dotted line at zero indicates time of contact. **c** Scatterplot showing the ratio of mean baseline firing rates between whisking and non-whisking epochs plotted against the ratios of evoked responses (max. firing rate in 10 ms following contact with baseline firing rate subtracted) between whisking and non-whisking ($W_{max}/NW_{max}$) for all Pr5 units. Same data shown binned as histograms. **d**, **e** Population PSTHs, identical to the one shown in **b** computed for all units recorded from the nucleus interpolaris (Sp5i) shown in green. Of these, only single-whisker-responsive single units were classified as putative interneurons (magenta). Darker colors denote non-whisking trials, whereas lighter colors denote whisking trials. **f** Scatterplot, identical to **c**, for all Sp5i units (green) with putative interneurons shown in magenta. Units having a baseline firing rate of zero during non-whisking epochs are shown at the top of the ordinate as having infinite $\overline{W}$ vs. $\overline{NW}$ ratios

(Supplementary Fig 3). Such cortex-lesioned animals were then implanted with identical electrode arrays as the intact animals and trained in the active touch task mentioned above. An example of a Pr5 recording in a lesioned animal is depicted in Fig. 3a. There were two differences in response profile when compared to the intact animal (cf. Fig. 2a). First, sensory gating was absent as can be clearly seen from equal peak response magnitudes during non-whisking (dark color) and whisking (light color) epochs. Second, the response was much narrower with only a single peak occurring within the first 2 ms post stimulus, giving rise to a much shorter response. There was a steep fall in rate after the peak followed by occasional spikes up to 20 ms post stimulus. Population data of these two effects in mean baseline-normalized, spike rates are plotted in Fig. 3b.

Ratios of peak response amplitudes and non-normalized, mean baseline firing rates were again computed for whisking and non-whisking trials for all units in the lesioned animals and are plotted in Fig. 3c (red) superimposed on the data from intact animals for comparison (blue, same as in Fig. 2c). The abolition of sensory gating is evident by the distribution of the $W_{max}$ vs. $NW_{max}$ ratios, which were now centered around 1 ($\mu$: 1.1, $\sigma$: 0.6, effect

size: 0.51, $n = 53$, $p = 0.35$, Wilcoxon's signed-rank test), whereas the baseline firing rates during whisking epochs remained elevated ($\mu$: 2.6, $\sigma$: 2, effect size: 0.38, $n = 53$, $p < 0.01$, Wilcoxon's signed-rank test). The distribution of evoked firing rates was significantly different between the intact and lesioned animals (effect size: 0.79, independent groups with $n = 23$ (intact animals) and $n = 53$ (lesioned animals), $p < 0.01$, Wilcoxon's rank sum test), whereas the baseline firing rate ratios remained unchanged (effect size 0.61, independent groups with $n = 23$ (intact animals) and $n = 53$ (lesioned animals), $p = 0.46$, Wilcoxon's rank sum test).

Next, response widths were measured by computing the effect sizes for each post-contact bin against the mean pre-contact baseline firing rate across units for intact (blue) and lesioned (red) animals, separately. Histograms of these effect sizes with statistically significant bins (green dots, $n = 23$ (intact animals) and $n = 53$ (lesioned animals), $p < 0.05$, Wilcoxon's rank sum test) are shown in Fig. 3d. The effect of the narrowing peak can be clearly seen by comparing the number of statistically significant bins exceeding 0.5 (no effect) across the two histograms.

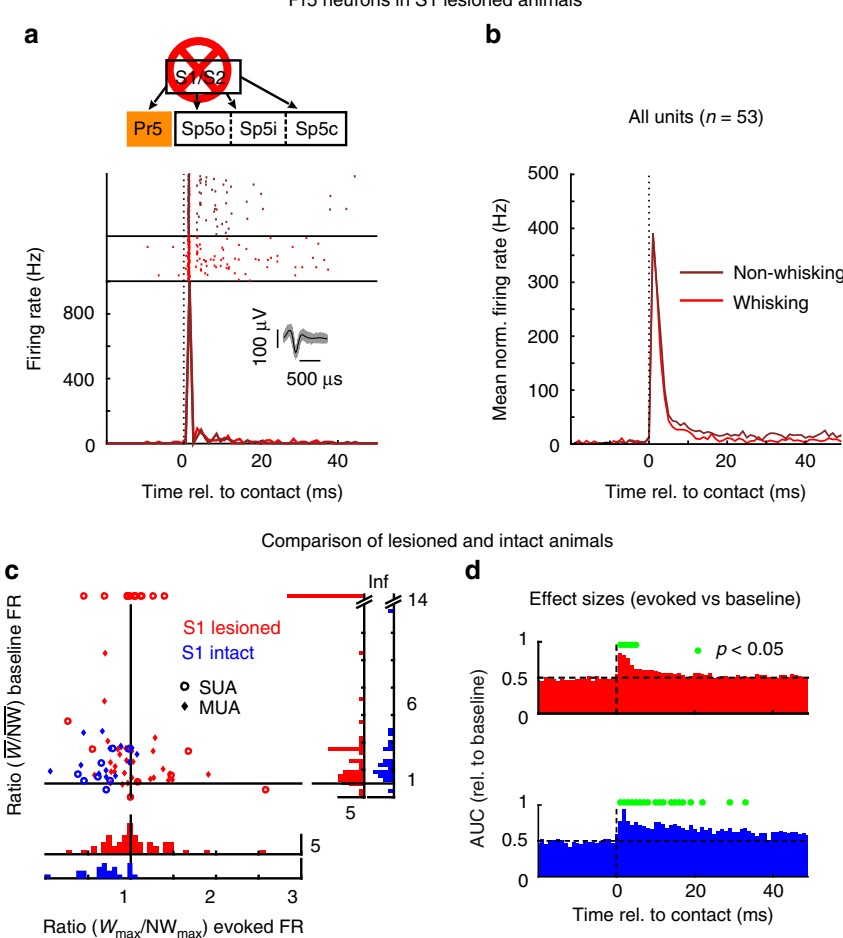

**Fig. 3** Sensory gating is cortex-dependent. **a** Representative example showing contact-evoked responses during non-whisking (dark red) and whisking (light red) epochs in a Pr5 neuron recorded in an animal with somatosensory cortex lesions. All conventions as in Fig. 2a. **b** Population PSTHs of all Pr5 units recorded from lesioned animals shown normalized to baseline firing rates as in Fig. 2b. **c** Scatterplot identical to that shown in Fig. 2c (blue), but now with the data from Pr5 neurons in lesioned animals plotted on top (red). **d** Comparison of peak widths of evoked responses in all Pr5 units in lesioned (red) and intact (blue) animals. Effect sizes (area under ROC curve; AUC) were calculated considering distributions of firing rates obtained for each post-stimulus bin vs. the mean pre-contact baseline ($n = 53$, lesioned; $n = 23$, intact). Bins that were significantly different ($p < 0.05$; Wilcoxon rank sum test) from baseline are indicated by a green dot

**Whisking kinematics and sensory gating.** Before concluding that the differences of spiking between whisking and non-whisking contacts and the lack of it in lesioned animals was due to sensory gating, we needed to exclude that the results could be explained by systematically biased whisking kinematics. To this end, we compared distributions of whisking kinematic parameters (set point, acceleration, and contact-induced deceleration) between whisking and non-whisking trials in intact and lesioned animals.

We first approximated the average protraction level by whisker position at contact during whisking and non-whisking trials and found neither a systematic bias between the two classes of contacts nor a systematic relation between contact-evoked firing and position that could explain the difference in evoked spiking (Supplementary Fig 4A, B).

Second, we found that evoked spiking is significantly correlated with contact strength approximated either by pre-contact acceleration of the whisker or by its deceleration when contact was established (Supplementary Fig 4C). As expected, whisking movements led to stronger hits (Supplementary Fig 4D). In contrast, our main result was an average attenuation of evoked responses in whisking trials (Fig. 2)—the opposite of the

expectation based on hit strengths. It is therefore difficult to align our finding with the idea that differing hit strengths are the basis of the difference found between whisking and non-whisking trials. This finding corroborates earlier studies, which excluded effects of hit strengths by using electrical stimulation of primary afferents and readily found sensory gating[9,10].

Third, we found only a slight difference in whisking kinematics in intact and lesioned animals (Supplementary Fig 5). In fact, lesioned animals even showed somewhat increased whisking amplitudes (effect size lesion vs. intact: 0.33), a finding that contradicts the notion that S1 is involved in generating elemental whisking movements[37]. Further, kinematic parameters related to hit strength from lesioned and intact animals for both whisking and non-whisking trials showed largely overlapping distributions and very similar differences between whisking and non-whisking pre-contact accelerations (Supplementary Fig 4D). Together, these findings exclude the possibility that a bias in hit strengths could have been at the base of the abolishment of the difference in contact-evoked spiking after S1/S2 lesions.

**Corticofugal activity: whisking- and contact-related signals.** The unexpected decrease in response width of Pr5 units due to

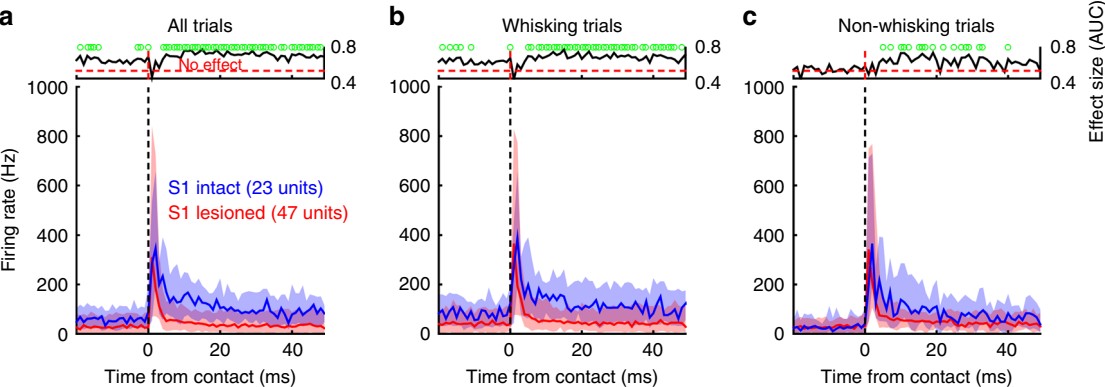

**Fig. 4** Corticofugal activity can be divided into whisking- and contact-related signals. **a** Medians (thick lines) and interquartile ranges (shaded regions) of contact-evoked Pr5 neuronal firing rates (without normalization) for all units recorded from lesioned (red) and intact (blue) animals. Only units with non-zero baseline firing rates were pre-selected for this analysis ($n = 47$, lesioned; $n = 23$, intact). Effect sizes (AUC) comparing pairs of bins at corresponding times between the two groups of animals are shown in inset (black line). Bins showing a significant difference between intact and lesioned animals are denoted by green dots. Vertical dashed lines show time zero (contact), whereas horizontal dashed line in inset denotes an effect size of 0.5 (no effect). **b, c** Identical PSTHs and effect size plots computed separately for whisking (**b**) and non-whisking trials (**c**)

somatosensory cortex lesions (Fig. 3d) demonstrates a robust modulation of ascending sensory activity by top-down, cortico-fugal projections[27]. Are these corticofugal modulations limited to specific epochs or do they result in a general decrease in response amplitudes? To determine this, we next quantified the temporal distribution of statistically significant, firing rate changes between intact and lesioned animals for all units with non-zero baseline firing rates in both whisking and non-whisking epochs ($n = 23$ units in intact and 47 units in lesioned animals). Figure 4a shows non-normalized, population PSTHs (medians with interquartile ranges) recorded from Pr5 units for all trials (whisking and non-whisking combined) in lesioned (red) and intact (blue) animals. Effect sizes (area under the receiver-operating characteristic (ROC) curve; AUC) were computed for each bin between the two groups (top inset) with statistically significant bins ($p < 0.05$, Wilcoxon's rank sum test) indicated with a green circle. Corti-cofugal modulations had significant effects on both evoked as well as baseline Pr5 activity.

To determine the origin of the increased baseline activity in intact animals, we next split the PSTH in Fig. 4a into two separate PSTHs for whisking (Fig. 4b) and non-whisking (Fig. 4c) trials. A striking difference between the two was the increased baseline firing rate in intact animals seen only during whisking trials. Temporal distributions of effect sizes for each trial type corroborated this with pre-contact baseline activity in non-whisking trials rarely exceeding an effect size of 0.5. Thus, corticofugal modulation of Pr5 activity manifested itself in two forms—an increased pre-contact, baseline firing rate, which only occurred during whisking (whisking-related) and a more prolonged, post-contact evoked response (contact-related), which was seen in all trials, as shown by the response width histograms in Fig. 3d. However, despite these effects, the whisking-related signal was still present in lesioned animals as demonstrated by the fact that the $(\overline{W/NW})$ distribution was centered above 1 (Fig. 3c). Therefore, whisking-related signals must come from multiple origins, one of them being cortex.

Given that both sensory gating and whisking-related signals were affected by cortical lesions, we also quantified the dependence of sensory gating on whisking kinematic parameters. As shown in Supplementary Fig 6, we neither found any dependence of gating on pre-contact, mean absolute whisking accelerations or velocities in a trial-by-trial fashion (Supplementary Fig 6A- D) corroborating earlier findings in S1[9].

**Whisking-related signals are not movement initiation signals**. The decline in whisking-related Pr5 neural activity upon removal of corticofugal input raises the important question about the functional implications of the signals carried by these pathways. One interesting possibility is that they convey information about movements used for sensory gating[31]. To determine the nature of the corticofugal signal and to test whether it contained a motor component, we used epochs where the animal went from rest to robust whisking without object contacts. For this, ratios of mean absolute velocities in the second and first halves of a sliding window moved across the whisking trace, were used to extract those instances where there was a transition from rest to max-imum velocity at the window midpoint[38]. The resulting velocity traces for intact and lesioned animals are shown in Fig. 5a, b, respectively with time zero indicating whisking onset. Figure 5c, d depicts the baseline-normalized firing rates of single and multi units (with at least 10 spikes in the pre-whisking period) triggered on whisking onset. Pr5 spiking increased following whisking onset. We next calculated the effect sizes between the firing rates during the 250 ms immediately preceding and following whisking onset (first and second halves of the window), for each neuron. The median effect size was 0.6 for intact animals and 0.64 for lesioned animals. Both groups showed a significant increase in firing rates (median: 13.6 Hz, interquartile range: 8.5–46.9 Hz, intact: $n = 17$; lesioned: $n = 33$, both $p < 0.01$, Wilcoxon's signed-rank test).

It is noteworthy that the firing rate increment occurred strictly after movement onset in both groups. It therefore cannot have played a role in movement initiation. Here, we term this a whisking-related signal to emphasize that it may convey sensory as well as motor aspects related to whisker movement[34,39,40], rather than a movement initiation signal.

## Discussion

In this study, we, for the first time, demonstrate the presence of sensory gating in TSN, the first synaptic stage of the whisker lemniscal pathway. We further demonstrate that it is pathway-specific and absent in the extralemniscal tactile stream. Finally, we provide evidence that gating is top-down in nature and abolished after lesions encompassing S1, S2, and surrounding parietal cortex. The function of the corticofugal signal is truly modulatory: whisking-related and contact-related signals exist without it, but

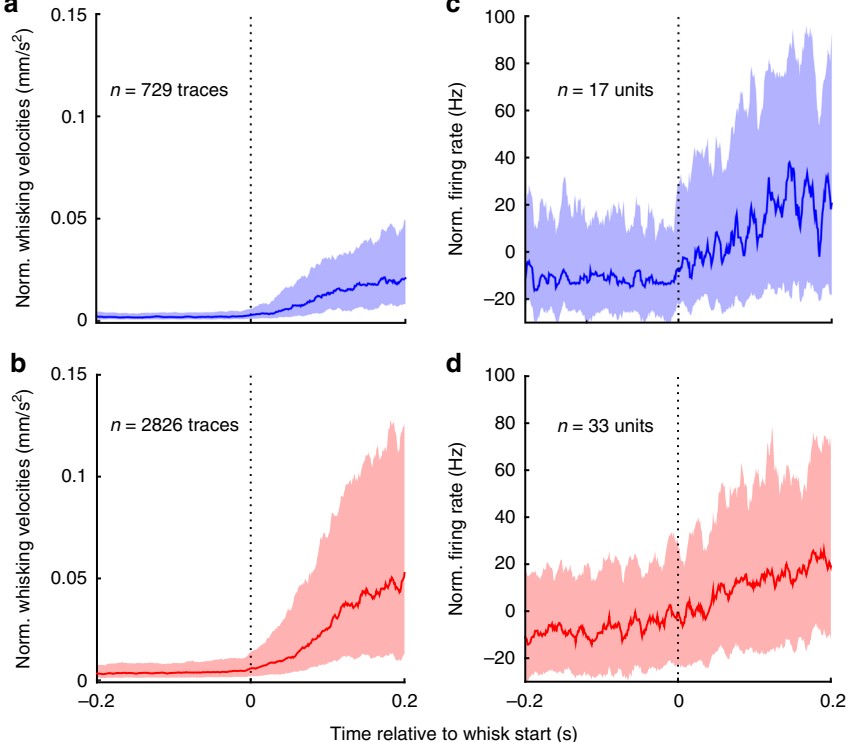

**Fig. 5** Whisking-related signals are not motor commands. **a**, **b** Median (thick line) and interquartile range (shaded) of whisking velocities computed for time windows in which animals initiated a whisk from rest at time zero for intact (blue, $n = 729$ traces) and lesioned (red, $n = 2826$ traces) animals. **c**, **d** Whisking-triggered PSTHs for all intact ($n = 17$) and lesioned ($n = 33$) Pr5 units (min 10 baseline spikes) for the same windows shown in **a**

are modulated by it in opposite directions, enhancing the first and suppressing the second.

Our data show that gating already occurs at the first station of the whisker-related tactile pathway. Sensory gating is a well-corroborated but highly varied phenomenon reported in different sensory systems, including the tactile system of rodents, cats, macaque monkeys, and humans[4–14,29,41]. While auditory and visual systems show gating only at thalamocortical stages of sensory pathways[41,42], the gating in the arm sensorimotor system of primates[11,12] and in the rat whisker system[10] was assumed to occur at earlier stations (but see conflicting results of ref. [7]). Seki and co-authors found evidence that gating involves primary afferent depolarization (PAD), a synaptic inhibitory mechanism believed to operate on primary afferents[36], Lee et al. demonstrated gating of electrically evoked stimuli in tactile thalamus only with microstimulation in tracts containing primary afferents, but not with stimulation of TSN fibers in the lemniscal tract[10].

Our Pr5 recordings confirm the notion of early gating in the tactile system by directly demonstrating its presence in Pr5. In rats, sensory gating might act via PAD as well, although, the existence of a PAD-based mechanism is uncertain. The synaptic elements potentially suitable for PAD have been demonstrated[43], but neither have their functional properties been shown to be suited for PAD nor have their origins been delineated. Pr5 itself contains only a small number of inhibitory neurons[32], and our finding of a lack of gating and motor-related responses in Sp5i putative inhibitory neurons does not support a mechanism based on an inhibitory cascade involving Sp5i[28], although these projections could still play a potential role, e.g. via axo-axonal synapses under cortical influence. We cannot exclude that other indirect projections of S1/S2 into Pr5 play a role. Many unexplored inhibitory brainstem targets of corticofugal projections may in principle be involved, amongst them Sp5c, which has been reported to project also directly to Pr5[32]. Other than PAD,

mechanisms such as postsynaptic modulation might also play a role particularly given that primary afferents terminate on distal dendrites and spines of Pr5 neurons[43]. Finally, corticofugal projections to Pr5[27] may act directly (e.g. via dendritic shunting mechanisms) on Pr5 dendrites to realize gating.

Studies in the primate somatosensory system have shown that gating is not a global downregulation of ascending sensory traffic but a remarkably refined and pathway-specific phenomenon that selectively affects only particular ascending sensory channels[29]. Whether sensory gating in the whisker system has similar properties was unknown as gating had only been studied in the thalamic and cortical stations of the lemniscal pathway[9,10]. Here, using recordings from both Sp5i and Pr5, we show that sensory gating in the rodent whisker system is also pathway-specific being only observed in Pr5, which gives rise to the lemniscal projections[44] and not in the caudal Sp5i where the extralemniscal projections originate[21]. Additionally, the Pr5 and Sp5i responses have different characteristics with the Sp5i responses being narrower and lacking the late component (>20 ms, cf Fig. 2b, d), which was shown to be dependent on corticofugal projections. These differences could reflect two possibilities. First, separate classes of corticofugal neurons projecting to the two separate nuclei could have different effects[45,46], the Pr5 projecting cells being responsible for the late response component mentioned. Second, sensory gating might be qualitatively different to the reafference principle by which sensory signals arising from ego motion are canceled out (see below).

In the tactile system, an important debate centers around the question of whether movement-related gating signals are conveyed to the sensory pathways via higher brain centers in a top-down fashion or whether they arise from proprioceptive signals. Sensory gating during passive movements in macaques[6,7] and humans[4] is compatible with the latter scheme, whereas the fact that gating survives peripheral deafferentation[4,9] requires a

central origin of gating signals. The strong modulatory influences of cortical stimulation on sensory signal flow not only supported a central origin of gating signals but raised the possibility that corticofugal signals might be involved[30,31]. We performed the critical experiment, chronically removing cortical drive during active sensation, while assessing gating in Pr5. Our lesions were large, encompassing S1/S2, which are the main sources of the cortico-trigeminal projection[27,28,33,47], but also including neighboring strips of parietal cortex. Histological analysis, however, precluded any possible inclusion of motor cortical areas. Thus, we clarify that without parietal cortex centered on areas S1/S2, peripheral gating breaks down. Our choice of the classical method of chronic lesioning over pharmacological or optogenetic blockade provides strong arguments in favor of the hypothesis that S1/S2 and neighboring areas are the exclusive origin and are critical for sensory gating. We found that gating was still abolished up to 3 months after lesioning (when the last recordings were made) thus safely excluding first, that the lost function is due to sudden withdrawal of homeostatic or balancing activity[48], and second, that other brain areas are capable of taking over the gating function based on plastic remodeling[49]. Further, the fact that whisking movements were unimpaired by the lesions (and even grew stronger on average) excludes the possibility that effects on movement generation led to the absence of sensory gating. Finally, we showed that the abolition of gating also reduces whisking-related signals in Pr5. Certainly, this does not prove a causal role for such signals in gating, but it is a possibility given that such whisking-related signals have been demonstrated in S1 of rats and mice[9,14]. It is worthwhile to mention here that a small part of the whisking-related signal was found to persist in the absence of corticofugal connectivity, which could either reflect sensory feedback from the periphery[34,39,40], or remaining top-down influences, for example, indirectly from motor cortex[28,38,50] although this motor-related activity was evidently not enough to sustain sensory gating.

How is sensory gating related to the hypothesis of the reafference principle[51–53], also sometimes referred to as state estimation[54,55]? In this view, a motor signal is used to predict the anticipated sensory consequences of an impending movement. Comparison of the motor-based prediction with the actual sensory feedback then results either in an improved estimate of the state of the motor plant and/or can be used to cancel reafferent, ego motion effects in sensory signals to improve perception[56–62]. Despite several outward similarities with sensory gating, such as its dependence on movement and its attenuation of the sensory stream, it is not completely clear whether reafference cancellation and sensory gating are the same phenomenon. In fact, our present results revealed important properties of sensory gating, which might not compatible with the notion of reafference cancellation in a parsimonious way. Here we speculate on the possibility that these two phenomena could reflect different functional mechanisms.

First, we found gating in the first neuronal station of the ascending tactile pathway—an unlikely place to encounter prediction signals. The generation of reafference prediction signals is believed to require complex neuronal operations, which, in mammals, are thought to involve cerebellum or cerebellum-like structures[56,57,63], from which the predictive signals are either conveyed to motor structures[64] or upstream to perceptual centers like the parietal cortex[65,66]. A cerebellar projection to Pr5 that could convey prediction signals to the tactile pathway in rat brainstem does not exist[67], and would be incompatible with our present finding that S1/S2 and neighboring parietal areas provide the gating signal. Thus, if reafference predictive signals cause sensory gating, S1/S2 would have to essentially calculate the cancellation signal and transmit it downstream via corticofugal

projections, a mechanistic element that needs further explanation in the framework of state estimation, as successfully corrected sensory signals would be expected to be rather conveyed upward toward higher cortical perceptual centers[54,63,68]. Second, an accurate predictive signal would subtract out ego motion from the sensory signal stream completely, so that the sensory responses to identical stimuli during active sensing and rest would be identical[56,57,69]. In contrast, our results, confirming the ones obtained on higher stations of the tactile pathway[8–10,14], demonstrate that tactile signals are diminished in amplitude during movement, an effect that deviates from the expectation that the purpose of this interaction is the stabilization of sensory signals originating from the outside world. Moreover, the magnitude of gating in rodents and monkeys appears to be independent of movement parameters[7,9] (Supplementary Fig 4), a finding that is not easily reconciled with the notion of correcting for movement-related consequences.

Perhaps the peculiarities of neuronal coding in the tactile system could explain these anomalies. A common purpose of whisking is texture identification, and in this context, leads to highly nonlinear and stochastic pattern of stick-slip events as the whisker interacts with a multitude of frictional and other forces[15,70]. Importantly, these events, while potentially encoded in unique neuronal activity that drive the animals' perception, are also susceptible to ego motion[71,72]. In this view, predictive signals in the tactile system should correct for probabilistic events that carry a potentially nonlinear and dynamic mix of signals about ego motion and the world. This would explain its lack of correlation with simple movement kinematics or its imperfect attenuation of sensory responses in our specific experimental situation and would support the view that sensory gating reflects a form of reafference-based cancellation of ego motion.

Another speculation that goes beyond even more generalized ideas about state estimation is that sensory gating reflects perceptual/cognitive parameters, e.g. the saliency of a stimulus or reward contingencies. In this view sensory gating would be the expression of another class of predictive coding, instantiating a mechanism to control sensory flow already in the periphery according to higher behavioral predispositions of the individual.

## Methods

**Animal surgery**. All experimental and surgical procedures complied with German Law for the Protection of Animals and were approved by the Regierungspräsidium in Tübingen, Germany. Female Sprague-Dawley rats, aged 11–13 weeks were implanted with movable electrode arrays into the Pr5 or the Sp5i according to standard procedures established in the laboratory[73]. Briefly, animals were anesthetized with a mixture of an opiate (fentanyl, 0.005 mg kg$^{-1}$), a benzodiazepine (midazolam, 2 mg kg$^{-1}$), and a α2 adrenergic agonist (medetomidine, 0.15 mg kg$^{-1}$), and gaseous isoflurane (1%). Subsequent doses of anesthetics were administered during the course of the surgery so as not to evoke toe pinch reflexes. The animal's body temperature was monitored using a rectal probe and maintained at 37 °C using a homeothermic blanket (Harvard Apparatus, MA, USA) and ophthalmic ointment was applied to the cornea to prevent drying. After trepanation, a single recording electrode was lowered to approximately 7 mm using stereotaxic co-ordinates to access Pr5 (9–9.5 mm caudal, 3 mm lateral to bregma) or caudal nucleus interpolaris, Sp5i (13–13.5 mm caudal, 3 mm lateral to bregma). Once responses to whisker deflections were obtained, the recording electrode was removed and a moveable array with four electrodes in a 2 × 2 arrangement was lowered to this position. After confirming the presence of spikes responding to whisker deflection, the array and its connectors were embedded into a head cap cast from dental cement and anchored to the skull using skull screws. The rims of the head cap were carefully formed and smoothed to allow seamless attachment of the skin after suturing. The wound was treated with antibiotic ointment and anesthesia terminated with an antidote (naloxon, 0.048 mg kg$^{-1}$; flumazenil, 0.1 mg kg$^{-1}$; atipam, 0.3 mg kg$^{-1}$ subcutaneous (s.c.)). Postoperative analgesia was provided by carprofen (5 mg kg$^{-1}$ s.c. for 36 h) and oral antibiotics (Baytril, Bayer Healthcare, Leverkusen, Germany) were provided for 14 days (2.5% in 100 ml drinking water).

For lesions of sensorimotor cortex, the S1 and S2 cortical whisker representations were first mapped using a recording electrode, the dura mater over the region of interest removed, and the cortex in the region [0.5, −4.5] mm

anterio-posterior and [3.5, 8.5] mm lateral to bregma was aspirated using a pulled glass pipette attached to the house vacuum. Suction was stopped once the white matter of the corpus callosum and external capsule were seen so as to minimize damage to fibers of passage (Supplementary Fig 1). The dorso-ventral arrangement of the whisker rows in Pr5 meant that advancing the recording electrode yielded neurons with receptive fields centered on different whiskers during each experimental session. Therefore, large lesions were made in cortex so as to completely remove all whisker-related corticofugal input to Pr5. Anatomical tracing performed earlier had shown that the TSN receive corticofugal input only from S1 and S2[27] and therefore the risk of affecting other sources of cortical drive with larger lesions was minimal. We were careful to reach complete hemostasis by rinsing and filling the cavity with gel foam (B. Braun Melsungen AG, Germany). The animal was then sutured and returned to its home cage and analgesics and antibiotics provided as detailed above. In order to avoid complications from brain stability possibly induced by the lesion cavities, we waited a minimum period of 4 weeks to allow the tissue around lesions to consolidate and reorganize, before the animal underwent the array implantation surgery.

**Behavioral task**. The animals were housed in groups of two under an inverted 12 h light-dark cycle. During behavioral training and recordings, the animal was given access to water only in the recording chamber and allowed to earn water till satiated. Behavioral sessions with recordings were carried out 5 days a week and the animals were given access to water ad lib over the 2 remaining days.

The animals were trained on an active touch task[9] as depicted in Fig. 1a. The animals were head restrained using standard procedures[73] and placed next to a glass rod mounted onto a magnetic cartridge that was in turn attached using a movable metal arm to a galvanometer[74]. The animals' whiskers were trimmed down to about 1.5–2 cm bilaterally and one of them, ipsilateral to the trigeminal recording electrodes, was elongated by a further 1 cm by slipping on a light polyimide tube (diameter: 250 µm, length: 2–2.5 cm, weight: ~0.7 mg). Whereas elongating the whisker by means of this tube certainly changes kinematic parameters of whisking, we used identical measures in lesioned and intact animals to minimize such effects. Contacts of this tube-mounted whisker against the moving glass rod were detected by the magnetic cartridge, which converted minute deflections of the object into voltage transients. This voltage signal was high passed filtered at 500 Hz and an amplitude threshold applied to distinguish whisker contacts from self-motion and vibration of the object during movement. Movements of the whisker and the glass rod were monitored using two laser optical devices (LOD MX series, Metralight Inc., San Mateo, CA, USA) each having a spatial and temporal resolution of 0.4375 µm and 0.4 ms, respectively. We chose active touch using object palpation over electrical stimulation methods since the latter result in a non-specific, highly repetitive and unnatural stimulation of the whiskers and does not reflect normal behavioral conditions.

We used a slight modification of the task as compared to the earlier study (Hentschke et al.[9]) in that we structured the session into trials. This was done to better control the rat's impulsive licking, and thereby sample valid trials more effectively. Each trial started with the moving arm being positioned next to the animal's face to signal trial onset and being moved in the rostro-caudal plane parallel to the animals face and about 3 cm lateral to it with white noise low-pass filtered at 5 Hz by a third-order Butterworth filter (Fig. 1b). Each contact, whether during active whisking or generated by the moving object deflecting the stationary whisker (non-whisking), was indicated by an auditory cue and rewarded with a small drop of water delivered through a spout attached to a piezo element and located 4–5 mm away from the tip of the lower jaw. The analog signal from the piezo element was passed through a threshold discriminator to generate a digital lick signal. Any licks after trial onset and before object contact resulted in trial abortion. After object contact, water appeared automatically and was not lick-dependent. The animal was then allowed as many consummatory licks as required. The next trial would start after a minimum of 0.5 s and an obligatory pause of licking activity of 200 ms, a measure that structured the behavior of the animal and effectively reduced the animals' impulsiveness. To encourage them to actively whisk to initiate contact and to discourage a strategy where they would simply place their whiskers in the path of the moving object and be rewarded by passive contacts, the midpoint and range of the moving object were constantly adjusted by the experimenter during a session. The entire task including rod movement and water delivery were controlled using the MATLAB Simulink environment and was deployed on a real-time engine using the XPC toolbox (Mathworks, Natick, MA, USA, Ver. 2013b).

**Electrophysiological recordings and spike sorting**. Movable multielectrode arrays were used to collect electrophysiological data. Before the commencement of the experiments, the microdrive was advanced in steps of approximately 60 µm until a satisfactory extracellular signal was obtained. Upon reaching the ventral aspect of the nuclei, the microdrive was retracted back to the original depth before advancing once more again. After two complete penetrations through the nucleus the experiment was terminated. At each recording site the four channels were monitored using auditory feedback (loudspeakers) and the whiskers stimulated using a handheld probe to determine receptive fields of spikes. The whisker that was represented in the receptive field of at least one recording channel was selected as the active whisker and monitored as described above. Both single units

(see description of spike waveform separation below) as well as multi units are reported in this study. Amongst all Pr5 units, all except one unit had SW receptive fields. Sp5i units were recorded mainly in the caudal half, the origin of the extralemniscal pathway, and the site of a dense population of interneurons, which can be discriminated from projection neurons by the extent of their receptive field[20,32,35]. Using careful handheld probe mapping, we identified 31% of units as responding to a SW and 69% to more than one whisker. Amongst the SW-responsive units, only those further classified as single units were designated as "putative interneurons" (Fig. 2e, f).

Voltage traces were recorded at a sampling frequency of 40 kHz and a gain of 5000× using a 16-bit multichannel extracellular amplifier (Multichannel Systems, Reutlingen, Germany). Voltage traces were bandpass filtered (200–5000 Hz) and referenced against one of the recording channels. The head cap was painted with silver lacquer and grounded to shield the recordings against movement artifacts. Spikes were extracted from this signal using a user-defined amplitude threshold. Local minima (troughs) that exceeded this threshold were identified and a snippet of data (0.3 ms before trough and 0.8 ms after) extracted. For troughs occurring within 1 ms of each other, the less negative of the two were discarded to avoid multiple copies of waveforms with bimodal features. All waveforms that exceeded 10 times the standard deviation of the signal were defined as artifacts and removed. The extracted snippets of data were then subjected to a principal component analysis (PCA) and the first three principal components were plotted against each other and recording time. Clusters in PCA space were identified using the k-means clustering algorithm or manually and were tracked along time until they merged with each other or the noise cluster. The waveforms were classified as single units according to conservative criteria, described earlier[75]. Clusters that did not satisfy the criteria for single units were then defined as multi units. Both single and multi units have been grouped together and reported throughout the paper as "units" except in the case of putative Sp5i interneurons (Fig. 2e, f) where only single units were used. All spike-sorting analyses were done by custom-written software in the MATLAB environment (Mathworks, Ver. 2014b).

**Histology**. Following the end of the experiment, animals were deeply anesthetized and a current of 10 µA for 10 s passed through each of the recording electrodes to mark their positions using electrolytic lesions (Supplementary Fig 2). The animal was then administered a fatal dose of sodium pentobarbital (50–100 mg kg$^{-1}$) and transcardially perfused with isotonic saline followed by 4% paraformaldehyde and the brains stored in 4% paraformaldehyde containing 30% sucrose at 4 °C for 2–3 days before being sectioned on a freezing microtome at a thickness of 60 µm. The brainstem was removed by blocking in the horizontal plane below the cerebellum and in the coronal plane at the rostro-caudal level of visual cortex and sectioned at a thickness of 60 µm. The brainstem sections were processed for cytochrome oxidase[76,77]. The ipsilateral half of the cortex of the lesioned rats was additionally sectioned in the coronal plane and stained for hematoxylin and counterstained with eosin according to standard procedures to visualize neuronal degeneration. Sections were mounted on gelatin-coated slides, coverslipped, and mosaic photomicrographs obtained using an Axio Imager Z2 (Carl Zeiss Microscopy, Jena, Germany) fitted with a moveable stage.

**Behavioral measurements in cortically lesioned rats**. Cortical lesioned animals were monitored for the first 3 days post lesion to determine if there were gross behavioral deficits with relation to whisking. Large, seemingly unimpaired whisker movements contralateral to the cortical lesion, and in synchrony with the healthy side, were seen by gross visual inspection in all lesioned animals immediately after waking from anesthesia (within an hour or 2 after lesioning). To quantify changes in whisking, freely moving animals whisking in a cage, the floor of which was covered with black paper, were filmed using a camcorder (Sony DSC RX100M3) at 30 frames per second. Only episodes in which the rat's head was held in parallel to the floor, the walls of the cage not in reach of the whiskers, were selected for analysis. Lines were then drawn manually on each individual video frame to mark the position of one single whisker on each side of the snout and the inter-ocular midline joining the snout to the forehead. The angle between these lines was calculated on a frame-by-frame basis and interpolated by a factor of 100 to yield whisking traces, which were then used to calculate whisking frequency power spectra.

**Classification of whisker contacts**. Criteria used to classify whisker contacts as occurring during non-whisking or whisking periods in the behavioral sessions under head fixation were similar to those published previously[9]. Briefly, whisker traces were first low-pass filtered at 100 Hz using a Butterworth third-order filter. Contact-triggered whisker traces were then extracted using a time window extending 75 ms before and after each contact. Time windows that had lick responses or a second contact within this window were discarded from further analysis. Pre-contact mean whisker velocities (10 ms preceding whisker contact) were extracted by taking the first derivative of the position trace and binned. The resulting histogram (Fig. 1c) was used to classify contacts during whisking and non-whisking states. The histogram of pre-contact velocities was typically bimodal with a peak centered around zero (non-whisking contacts) and a second broader peak (whisking contacts). The peak closest to zero was fit with a Gaussian (yellow

area in Fig. 1c) and velocities falling within double the Gaussian's standard deviation were defined as non-whisking contacts whereas the rest were defined as whisking contacts. Additionally, the root mean square whisker velocity in the 75 ms prior to contact had to be below 0.001 m s$^{-1}$ for the contact to be classified as non-whisking. This was done to prevent the incorrect classification of whisking traces that by chance had their apex (velocities close to zero) in the 10 ms preceding contact. These criteria were effective in separating whisking contacts from non-whisking contacts, as seen in a depiction of whisker kinematics.

As expected from the relative velocities of the whisker and the object, whisking contacts were systematically stronger than those occurring during non-whisking states. This was reflected by stronger decelerations around object contact during whisking (Fig. 1d) and was taken as being proportional to the force of contact. To avoid systematic biases in the force of whisker deflection, the distribution of minimum peri-contact decelerations was assessed for all whisking and non-whisking contacts. Only those trials falling into overlapping regions of these distributions (for whisking vs. non-whisking) were selected for further analyses of sensory gating.

**Quantification of neural responses.** Following the classification of whisker contacts, spike timestamps within a peri-contact window [−20, 50 ms] were used to calculate spike rates (at a temporal resolution of 1 ms). Average baseline rates within a 10 ms window preceding contact [−20, −10 ms] were compared to the touch-evoked neural response in post-stimulus window [0, 10 ms]. These time windows were used across all analyses and will be referred to as "baseline" and "evoked" periods throughout the paper. Only units whose evoked firing rate exceeded the baseline rate by three standard deviations were chosen for further analysis. Baseline firing rates were computed separately for whisking and non-whisking contacts.

To quantify baseline and contact-evoked responses the ratio of mean baseline firing rates of whisking and non-whisking trials $(\overline{W}/\overline{NW})$ and the respective ratio of peak evoked responses ($W_{max}/NW_{max}$) were computed. The peaks of evoked firing were assessed after smoothing the evoked firing rate (corrected for baseline) using a boxcar window of 3 ms. Effect sizes throughout the paper were computed using the AUC[78] and the population mean firing rates across groups tested for statistical significance the Wilcoxon rank sum test. Means and standard deviations denoted by $\mu$ and $\sigma$, respectively, are reported wherever applicable. All statistical comparisons were done using non-parametric measures and $p$ values computed using the Wilcoxon tests.

To compute peak widths (Fig. 3d), effect sizes (see above) were computed in a bin-by-bin fashion by comparing firing rates in each post-contact bin against the mean baseline firing rate across all neurons. Statistical significance was tested again in a bin-by-bin manner. For effect sizes between intact and lesioned animals (Fig. 4), effect sizes were similarly computed for each bin by comparing all neurons from lesioned animals against those from intact animals.

**Computation of whisking-related signals.** Neural signals related to whisking were computed to quantify a "whisking-related" signal, if any, in the TSN[38]. Briefly, whisking traces from each trial were low-pass filtered as described above, median filtered with a filter width of 7.5 ms, and down-sampled to 1 kHz. Instances where the animal started to vigorously whisk from rest were then extracted from this dataset. A 500 ms sliding window with 75% overlap was used to extract snippets of whisking data and the ratio of mean absolute velocities between the second and first halves of each such snippet calculated. Windows where the ratio of mean absolute velocities exceeded the 95th percentile, and which additionally had a mean absolute velocity <0.1 m s$^{-1}$ during the first half were further selected. These windows represented the strongest increases in whisking velocities occurring at the window midpoint. Windows which contained licks or object contacts were discarded from this analysis. A minimum of 10 viable windows were required for a particular session to be included in this analysis. Thus, two sessions containing three units (one intact and two lesioned) were excluded.

Spike counts in these windows were then binned and whisking-triggered spike rates computed and normalized by subtracting the mean baseline firing rate. To quantify the effect of whisking onset on spiking activity, effect sizes were calculated between firing rates in the first and second halves for each neuron. A distribution of such effect sizes for the entire neuronal population was obtained.

**Analysis of whisking kinematic parameters.** Four different measures of whisking kinematic parameters were extracted from the data, whisker position at contact, pre-contact velocity [−20, 0 ms], pre-contact acceleration [−20, 0 ms], and contact-induced deceleration (minimum acceleration [−5, 5 ms]).

Whisker position at the time of contact was taken as a measure of whisking set point. Whisker positions were extracted for whisking and non-whisking trials separately. To compare whisker positions between whisking and non-whisking trials, all whisker positions within a session were normalized to the interquartile range of positions from whisking trials of that particular session (Supplementary Fig 4A). Additionally, to determine if the kinematic parameters were encoded by Pr5 spike counts, correlation coefficients were computed for spike counts and each of the above kinematic parameters on a trial-by-trial fashion for each neuron for whisking and non-whisking trials separately. The distributions of these correlation

coefficients were then tested to determine if they were statistically different from zero (no correlation) with a Wilcoxon's rank sum test (Supplementary Fig 4B-C).

Additionally, to determine if pre-contact velocities and accelerations had an effect on gating, sensory gating (ratio of evoked firing rates between whisking and non-whisking trials) was computed on a trial-by-trial basis (Supplementary Fig 6). For this, the total spike count in each whisking trial [0, 10 ms] was divided by the median spike count across all non-whisking trials for that session values <1 thus indicating sensory gating. Spearmann correlation coefficients were then calculated between the kinematic parameters and evoked firing rate ratios for each trial.

**Extraction of peak responses.** To determine if there are multiple peaks in the Pr5 responses, we extracted mean PSTHs (1 ms bins) for each neuron for all whisking and non-whisking trials separately. Each PSTH was smoothed with a boxcar window of 3 ms and subjected to a peak detection algorithm where the first two firing rate peaks after whisker contact were extracted if they exceeded the mean baseline firing rate by three standard deviations and were separated from each other by at least 3 ms. This procedure was performed on whisking and non-whisking PSTHs separately. If the peaks for the whisking and non-whisking PSTHs, thus obtained, differed by more than 5 ms from each other, they were rejected. A total of 10 neurons exhibited such double peaks (Supplementary Fig 2A). Sensory gating was then computed for each peak separately by dividing the whisking peak by the non-whisking peak (Supplementary Fig 2B).

**Statistical measures.** As effect sizes could not be pre-determined, sample sizes were not chosen using power calculations. Intact and control group assignments were not randomized and the experimenter was not blinded regarding placement of lesions. No animals were excluded from the analyses. Appropriate statistical measures were used with effect size (AUC) and statistical testing (Wilcoxon's rank sum or signed-rank test) assuming non-parametric conditions. Standard deviations and/or interquartile ranges were used as a measure of variability.

**Code availability.** All analyses were performed using custom-written software using the MATLAB 2018a and the SIMULINK 2013a environments from Mathworks and are available from the corresponding author upon reasonable request.

## Data availability
All data are available from the corresponding author upon reasonable request.

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

## Acknowledgements

This work was supported by grants of the Deutsche Forschungsgemeinschaft (DFG) CH 1232/1-1 (Eigene Stelle) to S.C. and SCHW 577-16-1 to C.S and by the Werner Reichardt Centre for Integrative Neuroscience (CIN) at the Eberhard Karls University of Tübingen. The CIN is an Excellence Cluster funded by the Deutsche Forschungsgemeinschaft (DFG) within the framework of the Excellence Initiative (EXC 307). The authors would like to thank Mrs. Ursula Pascht for fabrication of electrodes and other excellent technical assistance and Dr. Richard L Roth for comments on an earlier version of the manuscript.

## Author contributions

S.C. and C.S. conceived the project and designed the experiments; S.C. performed the experiments, collected, and analyzed the data; S.C. and C.S. wrote the manuscript.

## Additional information

**Competing interests:** The authors declare no competing interests.

