## [Peer Review File · Nature Communications]

Reviewers' comments:

Reviewer #1 (Remarks to the Author):

In this article, the authors report that active touch is associated with gating of vibrissa-evoked responses in the PrV, as compared with passive touch. Yet, gating is absent in the SPVI. Gating principally affects late responses after the initial peak. Lesion of the somatosensory cortical areas (S1 and S2) abolishes gating; passive and active contacts result in similar response magnitudes. Therefore it is concluded that the cortex is required to gate sensory responses during active touch. Furthermore the authors show that whisking-related gating in PrV does not arise from motor commands per se.

Comments

Although I don't like the idea of elongating a vibrissa, which changes its inertia, and therefore the firing of primary afferents (see Severson et al., 2017) the same procedure was used in normal and lesioned rats.

The authors should be congratulated for performing such difficult experiments, which involve recording unit activity in the brainstem in alert head-restrained rats.

The study was carefully done with adequate quantitation of the data.

Yet, the actual source and mechanism of gating during active touch remain unknown.

I wonder to what extent gating relates to the average protraction level (setpoint) of the vibrissae when contact occurs during whisking and during passive touch? Can the authors be more explicit on this issue?

Severson et al. (2017) Active touch and self-motion encoding by Merkel cell-associated afferents. *Neuron* 94, 666-676.

Reviewer #2 (Remarks to the Author):

In this manuscript, Chakrabarti and Schwarz provide evidence for sensory gating in the brainstem nuclei of the rodent whisker system. The authors used an active touch task in rats while recording from the trigeminal sensory nuclei and show a suppression of responses during whisking compared to non-whisking epochs. They find evidence for this gating in the lemniscal nucleus Pr5 but not in the caudal interpolaris (Sp5i). Finally cortical lesions appear to remove the observed sensory gating and produce changes in the overall response profile of Pr5 neurons. The authors thus argue for the cortical dependence of this gating.

The major problem for both key conclusions of the paper - the presence of sensory gating in the brainstem nuclei and its cortical origin - is a lack of precise quantification of whisker kinematics. It is possible that the differences between whisking and non-whisking responses are due to differences in whisker movement. Similarly the differences between intact and cortical-lesioned animals can partially reflect differences in whisking and whisker-object contact. The authors have taken a number of measures to minimize this issue. However, these measures are not conclusive.

"As reported previously whisking contacts had significantly stronger decelerations around object contact as can be expected from the relative motion of the whisker and the object (Figure 1D). To prevent systematic biases, only trials contained in overlapping parts of minimum deceleration distributions of whisking and non-whisking trials were selected for further analyses."

Although this selection seems justified, it is not going to completely remove the bias. In fact, this selection might introduce a bias by eliminating from the analysis the stronger whisking contacts with high deceleration.

The authors analyse the whisking after cortical lesions and rule out that a deficit in whisking is the cause for the disappearance of sensory gating. However, again this analysis does not exclude the possibility of differences in whisking profile at the time of the experiments.

The analysis and the interpretation of findings related to Figure 5 are confusing. The authors observe an increase in Pr5 spiking activity following whisking onset. It is not clear why the authors refer to this as "motor-related". Based on previous research, the abrupt movement of whiskers is expected to produce an increase in firing rate of some Pr5 neurons. It is also not clear to me how this finding is "in accordance with data shown in Figure 3C" (last paragraph of results). The authors should consider rewriting this section for better clarity.

Reviewer #3 (Remarks to the Author):

In this paper, authors trained rats to perform an active touch task using their whisker and recorded the activities of the trigeminal neurons that relaying a tactile signal to the lemniscal and paralemniscal ascending pathway. Touch related neural responses were compared between active- and passive touch of whiskers. They found the smaller response to the active touch (sensory gating). Authors argued that this is a selective process in the specific pathway based on their finding that the size of sensory gating was different between two routes. This sensory gating is suggested to be induced by the primary sensory cortex since it was abolished after physical lesioning them. Authors concluded the finding reflects the top-down, cortico-fugal modulation at the earliest stage of sensory processing. Recording from small brainstem nuclei in the small rodent is very hard, so data presented here is unique and worth documenting. Also, the neural mechanism underlying the sensory gating is a central issue in the field of motor control. Therefore, I believe this finding will make a significant contribution to the field. I hope the following comment is helpful to improve this paper.

Major comments

1. This is a follow-up study to their recent publication in the *Frontier in Neural Circuits* that describing an anatomical basis of cortico-TSN projection, but this background was introduced insufficient way. Please add this in Introduction (including their finding of little projection from M1), and then please state what was the unsolved issue that is addressing in this paper.
2. Although the Pr5 neurons showed two or more peaks (Fig.2A, line 15-17 in the introduction), the ground average exhibited only single dominant peak. In addition, very early peak (almost zero latency) in B were not exist in A. I guess this reflects two possibilities; 1) Neuron shown in A is not representative in this aspect, and 2) earliest response (0 to 3-4ms from the contact, monosynaptic?), but not the later peaks and activities, were consistent among recorded cells. If so, instead of setting arbitrary time-window of [0,10ms] from touch (page 30, line 3 from bottom), it would be more appropriate to evaluate and discuss the sensory gating for these components separately.
3. After S1-2 lesion, the width of the earlier peak of Pr5 getting wider (by inspecting difference between Fig.2B and 3B), and changed more like the one of Sp5i (Fig.2D). Does this suggest the lower excitability (thus larger jitter in response to touch) in the second order neurons? Also, the late, long-lasting response exist only in the intact Pr5, but not the lesioned Pr5 as well as the intact Sp5i. Please discuss the possible reason.
4. Authors found that the whisking kinematics won't affect the amount of sensory gating (sup Fig.3), although the lesion of S1-2 increased the whisking amplitude. This result, as authors imply, may support the main conclusion of this paper under the assumption that the amount of sensory gating is

correlated with the size of movement amplitude and speed. This has been shown in other preparation including human psychophysics (e.g., Schmidt RF et al., EBR, 1990 (PMID:2311708), but may not found in the rodent whisker system. One idea is to show the size dependency of sensory gating in the normal rats, by using similar analysis as in the sup. Fig.3. For example, if the movement size was NEGATIVELY correlated with the size of sensory gating in the whisker system, authors argument could not be justified.

5. Given the fact that Sp5i also receive inputs from S1/S2, I am wondering why sensory gating was found only in the Pr5? Do authors suggest the different classes of S1/S2 pyramidal neurons to each sub-nuclei? Alternatively, were the gating effects from S1/S2 masked by another source of inputs (e.g., re-afference?)

Other comments

Page 4, line 13. "second order Pr5 neurons." Recorded neurons were not identified as the second order. No interneurons in the Pr5?. If so, please state that with reference.

Page 4, lines 16-7. See general point 1.

Page 6, bottom to Page 7, line 1. Consider moving this sentence to figure legend.

Page 8, 3rd paragraph. See main point 4.

Page 10, lines 1-2. How were the prolonged- evoked response (contact related) were generated? Please explain.

Page 10, line 1-2 from the bottom. No pre-movement signal doesn't necessarily mean no motor command?

Page 10, line 1 from the bottom. "genuine" motor signal? Please explain explicitly.

Page 11, line 14-15. "characteristic of motor command". See above.

Page 12, line 7-10. Since the PAD is induced by GABAergic interneurons, inhibitory interneuron is essential. The point here would be if the inhibitory interneurons mentioned here could form the axo-axonal synapse to the afferent, and receive top-down input.

Page 13, line 17. However, the chronic lesioning has proven disadvantages since it could facilitate plasticity in the whole system that is connecting to the lesioned area. For example, it is difficult to rule out the possibility that the sensorimotor system related to whisking may be altered while the retraining process after S1/S2 lesion. This argument, therefore, should be toned down. Page 13, lines 3-6 from the bottom. See main point 4.

Page 14-15, final section. The discussion here is speculative and needs to be toned down. Authors did evaluate neither the S1/2 component nor the re-afference component in the TSN's activity directly and did not evaluate how they were canceled out at the TSN level. Suggested example, I think, is not strong enough to evaluate this hypothesis.

Figure 1. Yellow shade in C is almost invisible. Please use the different shade.

Figure 1. Insets in B is too small. Please also add X-Y calibration bar.

Fig.2. If the projection from S1/S2 to all four TSN subdivisions were already known, please introduce those background in the introduction. It would also be helpful to introduce the known function of each subdivision briefly.

Fig.2C. Hard to dissociate SUA and MUA. Please use different color/shape.

Fig.2F. Better to put legend for each dot (pink and green).

Fig.4. The latency of the first peak looks earlier in the S1 lesioned units (significant?). Don't you think this as facilitation?

Fig.5BD. Why are the baselines negative? Is there a transient suppression before start whisking?

Sup Fig.3. See main point 4.

Point by point rebuttal

Reviewer #1 (Remarks to the Author):

In this article, the authors report that active touch is associated with gating of vibrissa-evoked responses in the PrV, as compared with passive touch. Yet, gating is absent in the SPVI. Gating principally affects late responses after the initial peak. Lesion of the somatosensory cortical areas (S1 and S2) abolishes gating; passive and active contacts result in similar response magnitudes. Therefore, it is concluded that the cortex is required to gate sensory responses during active touch. Furthermore, the authors show that whisking-related gating in PrV does not arise from motor commands per se.

Comments

1. Although I don't like the idea of elongating a vibrissa, which changes its inertia, and therefore the firing of primary afferents (see Severson et al., 2017) the same procedure was used in normal and lesioned rats.

RESPONSE: We agree with the reviewer that whisking kinematics are likely changed by elongation of the vibrissae, but as the reviewer himself/herself points out whiskers in both intact and lesioned animals were elongated using identical procedures. It is noteworthy that sensory gating has been readily observed in S1 cortex and thalamus when the whiskers were stimulated using a cuff electrode on the sensory nerves in the absence of any elongation. Therefore, although the reviewers concern is a legitimate one, we feel confident to state that sensory gating is present in the whisker system in the absence of whisker elongation (Fanselow and Nicolelis, 1999; Hentschke et al., 2006; Lee et al., 2008). We have now included a sentence in the methods section acknowledging the concern of the reviewer (page 28, line 793-795) and cite the Severson study (ref 39, page 19, line 520).

2. The authors should be congratulated for performing such difficult experiments, which involve recording unit activity in the brainstem in alert head-restrained rats. The study was carefully done with adequate quantitation of the data. Yet, the actual source and mechanism of gating during active touch remain unknown.

RESPONSE: We thank the reviewer for his/her kind comments. In this study, we concentrated on testing the hypothesis that sensory gating in the brainstem is a cortically mediated phenomenon via corticofugal projections. Although the sensorimotor cortex mediates brainstem gating, we agree that the source of this gating signal and its nature (motor, cognitive?) remain to be elucidated. Future experiments should aim to reveal the source(s) of predictive signals in S1 with a focus on contributions from the cerebellum and prefrontal areas. These aims are, however, beyond the scope of the present paper.

However, our data clearly point to the sensorimotor cortex as a critical component and the final common pathway of the circuits that mediate movement-related attenuation of sensory signals. This now affords us the possibility of systematically testing putative brain regions that project to S1/S2/PPC. We argue that this is a major advance in our understanding of sensory gating and predict a broad interest in our study (page 16, lines 418-429).

3. I wonder to what extent gating relates to the average protraction level (setpoint) of the vibrissae when contact occurs during whisking and during passive touch? Can the authors be more explicit on this issue?

RESPONSE: The reviewer raises the issue of the setpoint (average protraction level) of the vibrissae and asks us to be more explicit on the issue. Presumably the reviewer is concerned that the animals might have adopted a strategy where they specifically and systematically sampled different positions during the whisking and non-whisking trials. For example, it is conceivable that during whisking trials, more anterior positions were sampled. This would lead to a systematic difference in whisker position between whisking and non-whisking trials and might influence firing rates in these conditions should set point be encoded in Pr5 spikes.

To address the reviewers concern we extracted the whisker position at contact as a measure of setpoint. We chose this method instead of calculating the center of the whisking envelope as in our experiments free whisking epochs in the absence of contacts or licks were extremely short (<100ms). We first empirically determined whether the whisking positions during whisking and non-whisking trials within each sessions were systematically different. For this, we normalized the whisker positions for the non-whisking trials to the interquartile range of whisker positions of the whisking trials as shown in Supplementary Figure 4A. Ranges of whisker positions during whisking and non-whisking trials were largely overlapping and did not reveal any discernable systematic bias. We next measured the correlation between Pr5 spike counts and whisker positions in a trial by trial fashion for whisking and non-whisking trials separately in both intact and lesioned animals. The distributions of these correlation coefficients (Supplementary Figure 4B) were centered around zero and were statistically non-significant from zero, demonstrating a lack of set point encoding by Pr5 spikes. Thus, we think it is not likely that whisker set points are a potential confound for our findings.

These findings are presented in a new substantial sub-section entitled 'whisking kinematics and sensory gating' in the Results section of the revised manuscript (page 9, line 204).

4. Severson et al. (2017) Active touch and self-motion encoding by Merkel cell-associated afferents. Neuron 94, 666-676.

RESPONSE: We have now included this citation in the revision (ref 39, page 19, line 520).

Reviewer #2 (Remarks to the Author)

In this manuscript, Chakrabarti and Schwarz provide evidence for sensory gating in the brainstem nuclei of the rodent whisker system. The authors used an active touch task in rats while recording from the trigeminal sensory nuclei and show a suppression of responses during whisking compared to non-whisking epochs. They find evidence for this gating in the lemniscal nucleus Pr5 but not in the caudal interpolaris (Sp5i). Finally cortical lesions appear to remove the observed sensory gating and produce changes in the overall response profile of Pr5 neurons. The authors thus argue for the cortical dependence of this gating.

1. The major problem for both key conclusions of the paper - the presence of sensory gating in the brainstem nuclei and its cortical origin - is a lack of precise quantification of whisker kinematics. It is possible that the differences between whisking and non-whisking responses are due to differences in whisker movement. Similarly the differences between intact and cortical-lesioned animals can partially

reflect differences in whisking and whisker-object contact. The authors have taken a number of measures to minimize this issue. However, these measures are not conclusive.

RESPONSE: The reviewer raises the concern that both our main conclusions regarding the presence of sensory gating and its cortical origin could be explained by a difference in whisking kinematics between the whisking and non-whisking trials and between the intact and lesioned animals. The reviewer specifically raises the issue of whisker object contact. This comment aligns with respective comments of the other two reviewers (R1, comment 3, R3, comment 4). We now quantified the kinematic parameters pre-contact velocity and acceleration, as well as peri-contact acceleration to address the combined concerns of the reviewers (Supplementary Figure 4). The main findings of this analysis are presented in a new sub-section entitled 'whisking kinematics and sensory gating' in the Results section of the revised manuscript (page 9, line 204)

By definition the pre-contact velocity is close to zero in non-whisking trials. It follows that also pre-acceleration should be smaller in non-whisking trials, which is the case (compare top to bottom panel in the center column of panels in Suppl. Figure 4D; AUC effect sizes [printed in blue and red for intact vs. lesioned animals] are smaller than 0.5). Not surprisingly deceleration shows the opposite tendency, because high acceleration before contact tends to be followed by more deceleration, i.e. more negative acceleration during the hit (AUC values larger than 0.5 [printed in blue and red for intact vs. lesioned animals] shown in the right column of panels in Suppl. Figure 4D).

As Pr5 spike counts are presumed to signal 'hit strength' in some way, we expected that spike counts should be positively correlated with pre-contact accelerations and negatively correlated with contact induced decelerations (bc. high deceleration is indicated by more negative acceleration). These expectations were met in the population of recorded neurons (Suppl. Figure 4C).

Combining these two insights (i.e. whisking leads to stronger hits and stronger hits evoke more spikes) we predict stronger responses during whisking. Importantly, as shown in the main figure 2ABC, this expectation is not met by our results. In fact, the opposite is the case: non-whisking trials (containing the weaker hits) yielded stronger spike responses, clearly violating the above predictions and suggesting that kinematic influences on spiking were overwritten by other factors. Very similar observations and explanations were offered by previous studies that investigated VPM thalamus and S1 cortex (Hentschke et al., 2006; Wilent and Contreras, 2004). In conclusion, we hold it unlikely that a simple effect of pre contact or contact force differences can explain our results.

We would like to further point out that in previous reports by ourselves as well as other laboratories (Fanselow and Nicolelis, 1999; Hentschke et al., 2006; Lee et al., 2008) sensory gating has been repeatedly shown using electrical stimulation of the sensory nerve, methods that are immune to differences in whisker kinematics. Moreover, survival of gating after complete deafferentation of the whisker pad (Hentschke et al., 2006) firmly supported the notion that sensory gating is a central phenomenon. Taken these previous results together with the new analyses presented in the revised manuscript, we are confident that sensory gating in our data cannot be explained by differences in whisker kinematics.

2. "As reported previously whisking contacts had significantly stronger decelerations around object contact as can be expected from the relative motion of the whisker and the object (Figure 1D). To prevent systematic biases, only trials contained in overlapping parts of minimum deceleration distributions of whisking and non-whisking trials were selected for further analyses."

Although this selection seems justified, it is not going to completely remove the bias. In fact, this selection might introduce a bias by eliminating from the analysis the stronger whisking contacts with high deceleration.

RESPONSE: We refer to our response to comment 1 of this reviewer. We show there that stronger hits during whisking trials would predict stronger spike responses, not weaker responses as observed. Second, we point out that electrical stimulation with identical parameters did not abolish gating as shown previously.

To address the reviewer's comments about selection biases removed/introduced by preselection, we now recompute our results without any preselection (Supplementary Figure 1). In fact, neither the effect sizes between whisking and non-whisking trials nor those between lesioned and intact animals changed much because of pre-selection. We now mention this explicitly in the revised manuscript (page 5, line 102-104)

3. The authors analyse the whisking after cortical lesions and rule out that a deficit in whisking is the cause for the disappearance of sensory gating. However, again this analysis does not exclude the possibility of differences in whisking profile at the time of the experiments.

RESPONSE: The data now presented in Suppl. Figure 4 rule out this possibility (please refer also to our response to comment 1). We extracted pre contact velocities and accelerations and contact induced decelerations for all trials recorded in intact and lesioned animals. The difference between intact and lesioned animals yielded negligible effect sizes (printed in black in Suppl. Figure 4D).

4. The analysis and the interpretation of findings related to Figure 5 are confusing. The authors observe an increase in Pr5 spiking activity following whisking onset. It is not clear why the authors refer to this as "motor-related". Based on previous research, the abrupt movement of whiskers is expected to produce an increase in firing rate of some Pr5 neurons. It is also not clear to me how this finding is "in accordance with data shown in Figure 3C" (last paragraph of results). The authors should consider rewriting this section for better clarity.

RESPONSE: The reviewer is certainly correct. 'Motor-related' could simply be 'sensory'! Our terminology aimed to differentiate the signal from a motor initiation command (which would precede whisking onset). As we do not know the origin of the signal, we now call it 'whisking-related'. This neutral term does not preemptively assign 'motor' or 'sensory' properties to the signal. Following the reviewer's suggestion, we clarify the difference to motor initiation, and explicitly spell out that it may well be sensory (page 11, line 282-285).

Reviewer #3 (Remarks to the Author):

In this paper, authors trained rats to perform an active touch task using their whisker and recorded the activities of the trigeminal neurons that relaying a tactile signal to the lemniscal and paralemniscal ascending pathway. Touch related neural responses were compared between active- and passive touch of whiskers. They found the smaller response to the active touch (sensory gating). Authors argued that this is a selective process in the specific pathway based on their finding that the size of sensory gating

was different between two routes. This sensory gating is suggested to be induced by the primary sensory cortex since it was abolished after physical lesioning them. Authors concluded the finding reflects the top-down, cortico-fugal modulation at the earliest stage of sensory processing. Recording from small brainstem nuclei in the small rodent is very hard, so data presented here is unique and worth documenting. Also, the neural mechanism underlying the sensory gating is a central issue in the field of motor control. Therefore, I believe this finding will make a significant contribution to the field. I hope the following comment is helpful to improve this paper.

Major comments

1. This is a follow-up study to their recent publication in the *Frontier in Neural Circuits* that describing an anatomical basis of cortico-TSN projection, but this background was introduced insufficient way. Please add this in Introduction (including their finding of little projection from M1), and then please state what was the unsolved issue that is addressing in this paper.

RESPONSE: We thank the reviewer for his/her suggestion. We now mention our previous paper and point to the finding that no direct M1 projections to the trigeminal sensory nuclei were found (Introduction, page 3, lines 58-61).

2. Although the Pr5 neurons showed two or more peaks (Fig.2A, line 15-17 in the introduction), the ground average exhibited only single dominant peak. In addition, very early peak (almost zero latency) in B were not exist in A. I guess this reflects two possibilities; 1) Neuron shown in A is not representative in this aspect, and 2) earliest response (0 to 3-4ms from the contact, monosynaptic?), but not the later peaks and activities, were consistent among recorded cells. If so, instead of setting arbitrary time-window of [0,10ms] from touch (page 30, line 3 from bottom), it would be more appropriate to evaluate and discuss the sensory gating for these components separately.

RESPONSE: Following the reviewer's suggestions, we have included new analyses (Supplementary Figure 2; discussed on page 5, lines 113-120). We first determined the number of response peaks in each neuron recorded (see methods, page 35, line 983, Extraction of peak responses). We next show that 10 of the 23 neurons show two peaks separated by at least 3ms. The latencies of these peaks in whisking and non-whisking trials are now shown in Supplementary Figure 2A. We determined sensory gating on each peak separately and showed that gating occurred on both peaks (effect size 0.56; comparison of whisking/non-whisking firing rate ratios between first and second peaks). For the main text (Figs 2-3), we therefore kept our strategy to analyze the strongest peak within the first 10 ms following whisker contact.

3. After S1-2 lesion, the width of the earlier peak of Pr5 getting wider (by inspecting difference between Fig.2B and 3B), and changed more like the one of Sp5i (Fig.2D). Does this suggest the lower excitability (thus larger jitter in response to touch) in the second order neurons? Also, the late, long-lasting response exist only in the intact Pr5, but not the lesioned Pr5 as well as the intact Sp5i. Please discuss the possible reason.

RESPONSE: Upon the reviewer's suggestion we inspected the peak widths of the individual neurons in the intact and lesioned animals. Although it might appear to the eye from the grand averages (in panels 2B and 3B) that there may be a widening of the very first peak, this is not confirmed by quantitative analyses. The analysis of peak widths (Figure 3D) shows that there are more bins in the PSTHs with a significant response in the intact animals than in the lesioned animals.

The reviewer is absolutely right in pointing out that the Pr5 lesioned response is similar to the Sp5i response in terms of peak width (data not shown). Further, the very late response (>20ms) only appears in the intact Pr5. Thus, both gating as well as this late response may be corticofugal contributions to Pr5 responses, which are absent in Sp5i. This could reflect, as the reviewer himself/herself suggests in point 5, fundamentally different projection neurons targeting Pr5 and Sp5i, or alternatively, an interaction of different effects in the Sp5i (e.g. effect of reafference principle vs. gating). We have included a paragraph in the discussion clearly stating these possibilities (page 13, line 333) and thank the reviewer for this excellent suggestion!

4. Authors found that the whisking kinematics won't affect the amount of sensory gating (sup Fig.3), although the lesion of S1-2 increased the whisking amplitude. This result, as authors imply, may support the main conclusion of this paper under the assumption that the amount of sensory gating is correlated with the size of movement amplitude and speed. This has been shown in other preparation including human psychophysics (e.g., Schmidt RF et al., EBR, 1990 (PMID:2311708), but may not found in the rodent whisker system. One idea is to show the size dependency of sensory gating in the normal rats, by using similar analysis as in the sup. Fig.3. For example, if the movement size was NEGATIVELY correlated with the size of sensory gating in the whisker system, authors argument could not be justified.

RESPONSE: Following the reviewers suggestion, we measured sensory gating as a function of kinematic parameters such as pre contact velocities and accelerations in both intact and lesioned animals (Supplementary Figure 6). We found very low correlations between these variables in both intact and lesioned animals (page 11, line 259-263).

This comment is closely related to comment 3 of Reviewer 1 and comment 1 of Reviewer 2. Please also refer to our responses there and the substantial new material added in Suppl. Fig. 4!

5. Given the fact that Sp5i also receive inputs from S1/S2, I am wondering why sensory gating was found only in the Pr5? Do authors suggest the different classes of S1/S2 pyramidal neurons to each sub-nuclei? Alternatively, were the gating effects from S1/S2 masked by another source of inputs (e.g., re-afference)?

RESPONSE: We have addressed this concern together with the reviewers comment 3 (see above).

Other comments

1. Page 4, line 13. "second order Pr5 neurons." Recorded neurons were not identified as the second order. No interneurons in the Pr5?. If so, please state that with reference.

RESPONSE: As the reviewer correctly points out, 'second order Pr5 neurons' was a speculation on our part and was removed. We have instead added the following line (page 4, line 83-85)

Given the very low number of inhibitory interneurons in the Pr5, these most likely represent second-order neurons and added a reference.

2. Page 4, lines 16-7. See general point 1.

RESPONSE: We have added a paragraph on page 3, lines 58-61 clearly stating the importance of our previous work and its relation to the current study.

3. Page 6, bottom to Page 7, line 1. Consider moving this sentence to figure legend.

RESPONSE: We have moved this sentence to the legend but retained it in the text to help the reader (page 7, lines 162-163).

4. Page 8, 3rd paragraph. See main point 4.

RESPONSE: We have included a new section entitled 'whisking kinematics and sensory gating' (page 9, line 204) and a new Supplementary Figure 6 following the reviewer's suggestion (see above).

5. Page 10, lines 1-2. How were the prolonged- evoked response (contact related) were generated? Please explain.

RESPONSE: The prolonged evoked response was clearly corticofugal (see response to main point 3). We have clearly spelled this out and indicated that the prolonged evoked response was post contact in the text and not to be confused with pre-contact effects (page 10, line 253).

6. Page 10, line 1-2 from the bottom. No pre-movement signal doesn't necessarily mean no motor command?

RESPONSE: We agree with the reviewer. We have removed this line and instead added a paragraph (page 11, lines 282-285) clearly stating these signals could represent motor as well as sensory information about whisker movements.

7. Page 10, line 1 from the bottom. "genuine" motor signal? Please explain explicitly.

RESPONSE: We agree that this term is confusing. We have replaced this with the more unambiguous 'movement initiation signal' in the section heading which clearly should precede movement onset.

8. Page 11, line 14-15. "characteristic of motor command". See above.

RESPONSE: We have replaced this with movement initiation signal (page 11, line 283)

9. Page 12, line 7-10. Since the PAD is induced by GABAergic interneurons, inhibitory interneuron is essential. The point here would be if the inhibitory interneurons mentioned here could form the axo-axonal synapse to the afferent, and receive top-down input.

RESPONSE: This is a possibility and we thank the reviewer for bringing it up. We have now included this in the text with the following sentence (page 12, lines 312-316).

Pr5 itself contains only a small number of inhibitory neurons, and our finding of a lack of gating and motor-related responses in Sp5i putative inhibitory neurons does not support a mechanism based on an inhibitory cascade involving Sp5i as cause for gating, although these projections could still play a potential role, via axo-axonal synapses under cortical influence.

10. Page 13, line 17. However, the chronic lesioning has proven disadvantages since it could facilitate plasticity in the whole system that is connecting to the lesioned area. For example, it is difficult to rule out the possibility that the sensorimotor system related to whisking may be altered while the retraining process after S1/S2 lesion. This argument, therefore, should be toned down. Page 13, lines 3-6 from the bottom. See main point 4.

RESPONSE: The reviewer is correct in pointing out that lesions could induce plasticity. Despite this possibility – we do not observe the re-emerging of gating in the lesioned animals, even months after the lesion. This provides the strongest possible evidence for a critical role of corticofugal projections in

sensory gating. We concede that plasticity may happen. However, if it happens, it is not critical to our conclusion as it does not alleviate the blockade of gating.

11. Page 14-15, final section. The discussion here is speculative and needs to be toned down. Authors did evaluate neither the S1/2 component nor the re-afference component in the TSN's activity directly and did not evaluate how they were canceled out at the TSN level. Suggested example, I think, is not strong enough to evaluate this hypothesis.

RESPONSE: We agree that this is entirely speculative. We have now added the following sentence (page 15, line 386-387)

Here, we speculate on the possibility that these two phenomena could reflect different but overlapping functional mechanisms.

We have further added the following sentence acknowledging that this suggestion has caveats (page 16, lines 416-419)

This would explain its lack of correlation with simple movement kinematics or its imperfect attenuation of sensory responses in our specific experimental situation and would support the view that sensory gating reflects a form of reafference based cancellation of ego motion.

Figure 1. Yellow shade in C is almost invisible. Please use the different shade.

RESPONSE: We have darkened the color of the yellow shade to make it more visible.

Figure 1. Insets in B is too small. Please also add X-Y calibration bar.

RESPONSE: We have enlarged insets, added scale bars and changed the colors to red to clearly show they relate to the insets.

Fig.2. If the projection from S1/S2 to all four TSN subdivisions were already known, please introduce those background in the introduction. It would also be helpful to introduce the known function of each subdivision briefly.

RESPONSE: We have stated this in the introduction (page 3, line 58). The functions of these subdivisions are largely unknown and very speculative and we would prefer not to comment on this as this is clearly beyond the scope of our manuscript.

Fig.2C. Hard to dissociate SUA and MUA. Please use different color/shape.

RESPONSE: We have changed the symbols for better visibility

Fig.2F. Better to put legend for each dot (pink and green).

RESPONSE: We have introduced a legend

Fig.4. The latency of the first peak looks earlier in the S1 lesioned units (significant?). Don't you think this as facilitation?

RESPONSE: As discussed in main point 3, the slight earlier appearance of the peak in the grand average seems not to be represented across neurons in a systematic fashion. Hence we have avoided commenting on this issue.

Fig.5BD. Why are the baselines negative? Is there a transient suppression before start whisking?

RESPONSE: This is a result of normalization procedures by subtracting the mean baseline firing rate. This has been clearly indicated now in the Methods section (page 34, line 957)

Sup Fig.3. See main point 4.

RESPONSE: We have replaced the original figure with the new Supplementary Figure 6 where we have incorporated the reviewer's suggestions.

Bibliography

Fanselow, E.E., and Nicolelis, M.A. (1999). Behavioral modulation of tactile responses in the rat somatosensory system. *J Neurosci* *19*, 7603–7616.

Hentschke, H., Haiss, F., and Schwarz, C. (2006). Central signals rapidly switch tactile processing in rat barrel cortex during whisker movements. *Cereb Cortex* *16*, 1142–1156.

Lee, S., Carvell, G.E., and Simons, D.J. (2008). Motor modulation of afferent somatosensory circuits. *Nat Neurosci* *11*, 1430–1438.

Wilent, W.B., and Contreras, D. (2004). Synaptic responses to whisker deflections in rat barrel cortex as a function of cortical layer and stimulus intensity. *J. Neurosci.* *24*, 3985–3998.

REVIEWERS' COMMENTS:

Reviewer #1 (Remarks to the Author):

The authors have carefully addressed my critical comments. I am fully satisfied with the new version of the paper.

Only one remark:

Line 53 of the Introduction: The paralemniscal pathway arises from the rostral division of subnucleus interpolaris, not from the caudal division, which gives rise to the extralemniscal pathway.

Reviewer #2 (Remarks to the Author):

In the revised manuscript, the authors have made useful quantifications of the whisker kinematics. I also understand the authors' argument that the differences in whisker kinematics cannot explain the modulations of the activity for whisking trials. Nevertheless, I still find those differences important in interpreting the main conclusions of the manuscript. In Figure 1D, one can observe clear differences in whisker position trajectory, velocity and acceleration profiles. It is difficult to discern to what extent the observed differences induced during whisking are due to sensory gain versus effects of the pre-contact movement and presumably also adaptation.

The authors refer to previous papers that used electrical stimulation. I agree that the authors' findings are consistent with those reported using electrical stimulation. Nevertheless, those previous publications do not directly help in interpreting the findings here. Given the advantages of such electrical stimulations, it is unclear to me why the authors did not use electrical stimulation here.

The authors have now used the term whisking-related instead of motor-related. Unfortunately, I still find the use of the term whisking-related confusing and the last section of the Results hard to understand.

Reviewer #3 (Remarks to the Author):

All of my previous points are now reflected appropriately in this revision. This manuscript improved significantly after this revision, and I have no more concerns.

Rebuttal

Reviewer #1 (Remarks to the Author):

The authors have carefully addressed my critical comments. I am fully satisfied with the new version of the paper. Only one remark: Line 53 of the Introduction: The paralemniscal pathway arises from the rostral division of subnucleus interparietalis, not from the caudal division, which gives rise to the extralemniscal pathway.

RESPONSE: First, we thank the Reviewer for his/her kind comments and are delighted that he/she is satisfied with our revision. The Reviewer is indeed right in pointing out that the paralemniscal pathway arises from the rostral and NOT the caudal subdivision of the interparietalis. We have now corrected this error (page 3, line 55) and thank the Reviewer for pointing it out.

Reviewer #2 (Remarks to the Author):

In the revised manuscript, the authors have made useful quantifications of the whisker kinematics. I also understand the authors' argument that the differences in whisker kinematics cannot explain the modulations of the activity for whisking trials. Nevertheless, I still find those differences important in interpreting the main conclusions of the manuscript. In Figure 1D, one can observe clear differences in whisker position trajectory, velocity and acceleration profiles. It is difficult to discern to what extent the observed differences induced during whisking are due to sensory gain versus effects of the pre-contact movement and presumably also adaptation.

RESPONSE: The Reviewer accepts our argument that the differences in pre-contact force or kinematics, although clearly having an effect on the neural responses, modulate them in a manner which would yield effects opposite to those we observe (stronger not weaker responses during whisking). However, the Reviewer still points out that these differences (Fig 1D) should be considered when interpreting the main conclusions. We agree with the Reviewer that there are indeed differences in acceleration and velocity, as would be expected when comparing trials with and without whisking. We have indeed analyzed the relationship both between spiking responses (Supplementary Fig 4) and sensory gating (Supplementary Fig 6) to these whisking kinematic features. Whereas the differences in velocity and acceleration, as already mentioned above would actually influence the neural responses in a manner opposite to what we observe and since we do not see a systematic dependence between sensory gating and kinematics, we argue that the differences in response magnitudes between whisking and non-whisking trials cannot be attributed to kinematic differences.

The authors refer to previous papers that used electrical stimulation. I agree that the authors' findings are consistent with those reported using electrical stimulation. Nevertheless, those previous publications do not directly help in interpreting the findings here. Given the advantages of such electrical stimulations, it is unclear to me why the authors did not use electrical stimulation here.

RESPONSE: The Reviewer is correct in pointing out that our results are consistent with those obtained using electrical stimulation. Although electrical stimulation delivers stimuli of identical strength in all trials, it involves a highly artificial and repetitive global activation of the whisker sensory apparatus, one which is very incompatible with the animals natural behavior. Palpation against an object, on the other

hand, is a natural behavior that the animals regularly engage in and therefore a far better stimulus to determine the effects of sensory gating during movement. However, as the Reviewer himself/herself points out, electrical stimulation of the whisker pad performed by us previously and others show that sensory gating is not related to stimulus strength, which corroborates our analysis of gating and kinematic parameters. We have now added the following sentence to the methods summarizing the disadvantages of electrical stimulation (page 19, line 489).

,We chose active touch using object palpation over electrical stimulation methods since the latter result in a non-specific, highly repetitive and unnatural stimulation of the whiskers and does not reflect normal behavioral conditions.'

The authors have now used the term whisking-related instead of motor-related. Unfortunately, I still find the use of the term whisking-related confusing and the last section of the Results hard to understand.

RESPONSE: We had originally used the term motor-related but then changed it to whisking-related to accommodate the Reviewers earlier suggestion. We understand the Reviewers concern with the term ,related'. The signal we allude to is present during movement but is not modulated before movement onset, as a classical motor signal should. The term whisking related avoids terming it a whisking signal, suggesting classical motor attributes but associates this signal with whisking onset as it is present only in whisking trials. As the Reviewer does not offer an alternative suggestion that encompasses these aspects, we respectfully disagree and keep with our decision to term it a whisking related signal. We have however, added a phrase to the last section of the Results that unambiguously state that the term whisking related seeks to differentiate this from a movement initiation signal (page 12, line 295).

Reviewer #3 (Remarks to the Author):

All of my previous points are now reflected appropriately in this revision. This manuscript improved significantly after this revision, and I have no more concerns.

RESPONSE: We are delighted that he/she is satisfied with our revision.